# A Biocompatible Liquid Pillar[n]arene-Based Drug Reservoir for Topical Drug Delivery

**DOI:** 10.3390/pharmaceutics14122621

**Published:** 2022-11-28

**Authors:** Yahan Zhang, Mengke Ma, Longming Chen, Xinbei Du, Zhao Meng, Han Zhang, Zhibing Zheng, Junyi Chen, Qingbin Meng

**Affiliations:** 1State Key Laboratory of Toxicology and Medical Countermeasures, Beijing Institute of Pharmacology and Toxicology, Beijing 100850, China; 2Key Laboratory of Inorganic-Organic Hybrid Functional Material Chemistry, Tianjin Key Laboratory of Structure and Performance for Functional Molecules, College of Chemistry, Ministry of Education, Tianjin Normal University, Tianjin 300387, China

**Keywords:** liquid pillar[n]arenes, econazole nitrate, drug reservoir, topical administration, sustained release

## Abstract

Advanced external preparations that possess a sustained-release effect and integrate few irritant elements are urgently needed to satisfy the special requirements of topical administration in the clinic. Here, a series of liquid pillar[n]arene-bearing varying-length oligoethylene oxide chains (OEPns) were designed and synthesized. Following rheological property and biocompatibility investigations, pillar[6]arene with triethylene oxide substituents (TEP6) with satisfactory cavity size were screened as optimal candidate compounds. Then, a supramolecular liquid reservoir was constructed from host–guest complexes between TEP6 and econazole nitrate (ECN), an external antimicrobial agent without additional solvents. In vitro drug-release studies revealed that complexation by TEP6 could regulate the release rate of ECN and afford effective cumulative amounts. In vivo pharmacodynamic studies confirmed the formation of a supramolecular liquid reservoir contributed to the accelerated healing rate of a *S. aureus*-infected mouse wound model. Overall, these findings have provided the first insights into the construction of a supramolecular liquid reservoir for topical administration.

## 1. Introduction

With the rapidly evolving development of medical care, improvement of patient compliance has generated extensive attention [1,2,3,4,5]. Since the innovation of topical administration, such a route has been widely employed in the clinic and is being increasingly considered as a promising alternative to oral or invasive administration due to its convenience and affordability [6,7,8,9,10]. Most available formulations, including creams, gels, ointments and other forms, show burst drug release, which requires repetitive daily application or high-dose administration, causing poor adherence, uncomfortable adverse reactions and high-financial burden [11,12,13]. One effective strategy to tackle this challenge is to form a drug reservoir on the skin to achieve suitable sustained release at the application site [14,15]. Different types of materials such as polymers, silicon and dendrimers have been applied to prolong drug release [16,17,18,19,20]. These materials tend to be blended with organic solvents or other excipients, which pose the potential for adverse effects, such as skin irritation and erythema at the application site [21,22,23]. It is of paramount importance to explore novel materials that are capable of constructing a drug reservoir for long-term drug release without barrier disruption or irritant additives.

A supramolecular system based on synthetic macrocycles has broad potential applications for biology and medicine by leveraging specific, directional and reversible non-covalent molecular recognition motifs [24,25,26,27,28]. One particular macrocycle, pillar[n]arenes [29,30] with intrinsic rigid architecture and easy modification have been considered as a promising candidate for drug carriers or therapeutic modifiers [31,32,33,34,35]. Compared to traditional biomaterials, the novel pillar[n]arene-based drug delivery system exhibited remarkable potential and advantages, such as a good loading capacity of its cavity, sustained and controlled delivery of therapeutic drugs due to reversible host–guest interactions [36,37]. A variety of elegant delivery systems have consecutively come out in the last 10 years, while to the best of our knowledge, no examples of pillar[n]arene-based materials serving as a drug reservoir for topical administration have been exploited. Therefore, we reported herein a series of liquid pillar[n]arene derivatives by selectively introducing oligoethylene oxide groups (OEPns), which were further applied to construct a supramolecular reservoir without the addition of solvents for drug sustained release, as schematically depicted in Figure 1. Among these, pillar[6]arene bearing with triethylene oxide substituents (TEP6) with fine rheological properties, excellent biocompatibility and satisfactory cavity size was screened as optimal macrocyclic host. Complexation of econazole nitrate (ECN) [38,39,40,41], an external antifungal agent with TEP6 could afford a suitable sustained release for 24 h compared to free ECN and the mixture of its monomer (M) and ECN in vitro. In addition, the drug reservoir ECN/TEP6 complex exhibited superior wound repaired efficacy on *S. aureus*-infected mouse wound model.

## 2. Materials and Methods

### 2.1. Materials and Physical Measurements

All agents and materials were of analytical grade without further purification. Econazole nitrate, di-, tri- and tetra-ethylene glycol monomethyl ether were purchased from Energy chemical. The ^1^H- and ^13^C-NMR spectra were recorded at Bruker AVANCE 600 MHz. Molecular weights were analyzed using matrix-assisted laser desorption/ionization time-of-flight mass spectrometry (MALDI-TOF-MS; Bruker Reex, Bruker, Billerica, MA, USA). SpectraMax^®^ M5 plate reader, (Molecular Devices) was used for cytotoxicity studies. Rheological studies were analyzed on an HR-10 rheometer (TA instruments). High-performance liquid chromatography (HPLC) analysis was performed by using an LC-20AT instrument configured with an SPD-20A detector and a C18 column (4.6 × 250 × 5 μm).

### 2.2. Rheological Studies

Continues flow was measured on a controlled-stress DHR-10 rheometer according to previously reported methods [42,43]. Parallel plate geometry with 20 mm diameter (sample gap of 200 μm) was introduced. Then, rheological properties were generated for each sample in a controlled shear rate ranging from 0.01 to 100 s^−1^ at 25 °C. Additionally, the same parallel plate was used for the determination of oscillatory rheometry. The storage (*G*′) and loss (*G*″) moduli were measured when the frequency sweep was performed over a range of 1 to 100 rad/s.

### 2.3. Biocompatibility Evaluation of OEPns

The relative cytotoxicity of the OEPns against the human-immortalized keratinocytes (HaCaT) cells (Chinese Academy of Science, Beijing, China) was determined using a CCK-8 assay [44]. HaCaT cells were cultivated into a 96-well cell culture plate at a density of 8000 cells per well in 100 μL of Dulbecco’s Modified Eagle medium (DMEM) (Gibco, Madrid, Spain) supplemented with 10% fetal bovine serum (FBS) (Invitrogen, Waltham, MA, USA), 1% penicillin (Invitrogen, CA, US) and 1% streptomycin (Invitrogen, CA, US). DEP5, DEP6, TEP5, TEP6, QEP5, QEP6 at different concentrations (5, 10, 20, 40, 80, 160, 320 μmol/L) were added to the cell-containing wells and incubated at 37 °C under humidified 5% CO_2_ for 48 h. Then, 10 μL of CCK-8 reagent and fresh cell culture medium were added into each well and incubated in the dark for 30 min. The absorbance was measured spectrophotometrically at 450 nm.

Kunming mice were purchased from SPF Biotechnology Co., Ltd., (Beijing, China) and divided into seven groups (*n* = 3). The back hair of all the mice was shaved and a circular spot with a diameter of 8 mm was delimited on the back skin. Each group of mice were evenly rubbed with DEP5 (10.40 mg), DEP6 (11.85 mg), TEP5 (13.21 mg), TEP6 (15.86 mg), QEP5 (16.03 mg) and QEP6 (19.22 mg) (all equivalent to 6.38 μmol TEP6) on the center of each spot once every two days. Normal mice were treated with PBS as a control group. All mice were sacrificed after 10 days. Skin tissue with the delimited spot was collected and stained with hematoxylin and eosin (H&E) for histopathological analysis.

### 2.4. NMR Spectroscopy

Samples of TEP6 (5 mM) and ECN (5 mM) for NMR measurements were prepared in CDCl_3_. Samples for 1:1 ^1^H-NMR spectra were prepared in CDCl_3_. All spectra were acquired at 298 K in the solution state.

### 2.5. Fluorescence Titration

To determine the association constant between TEP6 and ECN, a direct fluorescence titration of TEP6 (1 μM) with ECN was carried out in a methanol/water (70:30, *v*/*v*) solution according to previously reported methods [45].

### 2.6. In Vitro Permeability Studies

Abdominal skin (approximately 5 × 5 cm) was excised from the Kunming mice within the first hour of animal death. Once obtained, hairs were shaven and the epidermis was prepared by heating at 60 °C for 2 min and placed between the donor (up) and receptor (down) chambers of Franz-diffusion cells with an area of 1.77 cm^2^ available for diffusion. 15 mL methanol/water (70:30 *v*/*v*) [46,47] solution was added into each receptor chamber as receptor medium (37 °C and 250 rpm). In ECN group, the donor chambers were filled with ECN (22.5 μmol, 5.65 mg/cm^2^). ECN/TEP6 (ECN: 22.5 μmol, 5.65 mg/cm^2^; TEP6: 22.5 μmol, 31.59 mg/cm^2^) and M + ECN (ECN: 22.5 μmol, 5.65 mg/cm^2^; M: 135 μmol, 28.22 mg/cm^2^) were placed on the skin surface in the donor compartment in the TEP5 and M groups, respectively. 100 μL of samples were withdrawn at 0.5, 1, 2, 4, 6, 8, 16, and 24 h and replaced with fresh receptor solution. The calibration curve of ECN was derived by HPLC. The amount of ECN that permeated through the skin was measured and each result represents the mean value of three experiments.

### 2.7. In Vivo S. aureus-Infected Mouse Wound Model

The method for the establishment of the wound healing model used in this study was presented in a previous publication [48]. In brief, Kunming mice were anesthetized with isoflurane. Then, hairs of dorsal skin were shaven and an 8 mm diameter skin wound was created. Twelve mice were randomly divided into four groups (*n* = 3): control, ECN, M + ECN and ECN/TEP6. Subsequently, each wound was filled with a suspension of *S. aureus* (ATCC 25923) (Fuxiang, Shanghai, China) at a concentration of 1 × 10^6^ CFU. The uninfected group was not infected with *S. aureus*. After one day, the mice in the different groups were, respectively administrated with ECN (6.36 μmol, 5.65 mg/ cm^2^), M + ECN (a 1:6 mixture of ECN and M; ECN: 6.36 μmol, 5.65 mg/ cm^2^; M: 38.16 μmol, 28.22 mg/cm^2^), ECN/TEP6 (a 1:1 mixture of ECN and TEP6; ECN: 6.36 μmol, 5.65 mg/ cm^2^; TEP6: 6.36 μmol, 31.59 mg/cm^2^) and 30% DMSO. The mice were treated through topical rubbing onto the affected area every two days. The process of wound healing was regularly observed (0, 2, 4, 6, 8, and 10 days) and the wound area was recorded. In addition, on the 10th day, the skin tissue was fixed using formalin 10% *v*/*v*, and then incorporated into paraffin. Blocks of 4 μm thickness were obtained and stained with H&E.

### 2.8. Statistical Analysis

Quantitative data are reported as the mean ± standard deviation. Analysis of variance was performed by one-way analysis of variance (ANOVA) and Student’s *t* test.

## 3. Results

### 3.1. Synthesis and Characterization

Candidate macrocyclic compounds that could be used for the construction of a liquid supramolecular reservoir should be subtly designed considering the following features: (1) fine-flow properties to facilitate application; (2) excellent compatibility to avert skin disruption; (3) satisfactory binding affinities towards drug molecules; and (4) suitable sustained-release ability to optimize repeat administration. Pillar[n]arenes were employed as a macrocyclic scaffold benefiting from their distinctive recognition properties as well as facial modification. Oligoethylene oxide substitutions were attached to afford adequate flow properties [49,50] and biocompatibility [51,52]. Following the synthesis routes similar to those previously described [53,54], we synthesized six liquid pillar[n]arene derivatives used for subsequent investigations (see Appendix A for details of synthetic procedures and compound characterization).

### 3.2. Rheological Properties

Rheological properties of the OEPns were first assessed by a controlled-stress DHR-10 rheometer. As shown in Appendix A, the oscillatory shear viscoelastic response signals exhibited that the loss moduli (*G*″) of the OEPns were greater than their storage moduli (*G*′), proving that these liquid pillar[n]arene derivatives were predominantly viscous. Then, continuous rheological analysis revealed that QEP5 and QEP6 presented a linear relationship between shear stress and shear rate, and their viscosity was constant within the examination range, a typical feature of Newtonian fluid (Figure 1a). In addition, when shear rate was 1 s^−1^, shear viscosity of QEP5 and QEP6 was less than 2.0 Pa·s, indicating that these two compounds were inadequate for attachment to the skin surface [55] (Figure 1b). While the viscosities of TEP5, TEP6, DEP5 and DEP6 exhibited shear thinning, a non-Newtonian characteristic, in which viscosity reduced as the shear rate increased (Figure 1a). Additionally, at a shear rate of 1 s^−1^, the viscosity of these four derivatives was over 4.0 Pa·s, in favor of their uniform distribution in topical administration (Figure 1b).

### 3.3. Safety Profile of the OEPns

Prior to testing their efficacy as a topical drug reservoir, cytotoxicity of the OEPns on human keratinocyte cells (HaCaT) was evaluated using a Cell Counting Kit-8 (CCK-8) assay. DEP5 and DEP6 displayed concentration-dependent cell death. When the concentration reached 320 μM, the cell variability was quantitatively measured to be 34.60 (±4.63) and 25.97 (±4.32)%, respectively (Figure 2a,d). More than 95% of the cells survived within examination range after incubation with the other four compounds for 48 h, suggesting that the cytotoxicity was negligible (Figure 2b,c,e,f). Histological analysis of the skin tissue of the mice was performed to give further support for safety of OEPns (Appendix A). Discernable disruption of the stratum corneum and epidermis was observed in the DEP5- and DEP6-treated group, unlike the PBS assay control. Tissue slices of the other four compound-treated groups retained a normal histopathological morphology. Overall, considering both the rheological properties and biocompatibility, TEP5 and TEP6 were screened to be optimal candidate compounds for topical application.

### 3.4. Host–Guest Recognition

In view of the cavity size, pillar[6]arene (approximately 6.7 Å) might accommodate a greater variety of guests [56]. Therefore, we took TEP6 as a representative to conduct the following performance evaluations. Econazole nitrate is an external antimicrobial agent, market preparations of which are frequently administrated (twice a day), causing poor adherence in patients. Given its size complementarity with that of TEP6, ECN was selected as a model drug. Firstly, the complexation between TEP6 and ECN was investigated by ^1^H-NMR spectroscopy. Upon addition of one equivalent of TEP6, the protons of ECN, especially for H_a_, underwent a substantial upfield shift (Δδ = −0.25 ppm) and experienced significantly broadening effects compared with free ECN (Figure 3). Meanwhile, the proton signals for TEP6 shifted downfield, presumably as a consequence of the complexation-induced deshielding effect.

Having satisfied the most crucial parameters by ^1^H-NMR spectroscopy, we sought to evaluate the host–guest complexation between TEP6 and ECN quantitatively using fluorescence titration experiments. As shown in Figure 4a, along with the increasing concentration of ECN, the fluorescence intensity at 325 nm decreased gradually, which was ascribed to photo-induced electron transfer. According to a standard curve fitting protocol, the association constant (*K*a) for ECN with TEP6 was calculated as 5.37 ± 0.65 × 10^3^ M^−1^ (Figure 4b). In addition, the geometry optimization of ECN/TEP6 was performed using a MM2-minimized molecular model (Appendix A). The results verified the formation of a 1:1 complexation between ECN and TEP6.

### 3.5. In Vitro ECN Release Studies

As supported by above ^1^H-NMR spectrum and fluorescence titration results, TEP6 was expected to effectively bind to ECN in the absence of additional solvents. To the extent this supposition was correct, we envisioned that a mixture between TEP6 and ECN could be used to construct a supramolecular liquid reservoir capable of sustained release, thus allowing topical application. Here, a Franz-type diffusion cell system was used to evaluate the influence of forming a liquid reservoir on the drug release rate. An appropriate calibration curve of ECN was first derived by HPLC (Appendix A). The plot of the cumulative permeable amounts of several schemes as a function of time are depicted in Figure 5a. Free ECN exhibited a fast drug release within 8 h, and a similar tendency was also found in the M+ECN group. While co-application with TEP6 led to a significant sustained release and cumulative permeable concentrations reached approximately 70% over the course of 24 h. This outcome led us to suggest that the formation of a supramolecular liquid reservoir might effectively regulate the drug release rate, important in achieving an optimal clinical effect of the agents, reduce undesired side effects and improve patient compliance [57].

### 3.6. Wound Healing Performance

We then evaluated whether forming a supramolecular liquid reservoir could provide salutary benefits for the therapeutic effect of ECN. The *S. aureus*-infected mouse wound model was employed to evaluate the antimicrobial efficacy of ECN/TEP6 in vivo. In order to demonstrate the establishment of the infected model, the Un-inf group which represented mice with uninfected skin wounds was designed according to the reference [58]. As shown in Figure 5b,c, compared to the wound site of the Un-inf group on the 10th day, the infected wound site of the control group recovered slowly, and the wound size remained 19.07 ± 3.40 mm^2^ over a course of the 10 days, accompanied by redness and swelling. These results indicated that the infected model was established successfully. Unlike the control group, free ECN and M+ECN led to rapid wound healing and relieved symptoms to a certain extent. Notably, a large and statistically significant acceleration was observed in the wound recovery for the ECN/TEP6 group. On day 10, the average wound area reduced to only 2.09 ± 0.72 mm^2^.

Furthermore, H&E staining was carried out to assess the therapeutic efficiency of each group (Figure 5d). It was apparent that the epidermis structure of the wound sections of the control group thickened after the fall of eschar. In addition, hemorrhage, macrophage infiltration and collagen deposition were observed in the dermis. Compared to the control group, morphological assessment of the ECN and M + ECN groups revealed symptomatic relief to some degree. Remarkably, treatment with ECN/TEP6 exhibited markedly less tissue damage similar to that of the uninfected group as inferred from the corresponding histological analyses. On these basis, we proposed that the formation of a supramolecular liquid reservoir by ECN/TEP6 could achieve a sustained release of ECN while improving its antimicrobial efficacy.

## 4. Conclusions

In summary, we have synthesized six liquid pillar[n]arene derivatives by attaching varying-length oligoethylene oxide chains. Among these, TEP6 was screened as the optimal candidate compound according to its rheological properties, biocompatibility and cavity size. More significantly, an available supramolecular liquid drug reservoir that involves the direct mixture with a conventional antimicrobial agent ECN was successfully constructed without additional solvents and was applied for topical drug delivery. An in vitro diffusion cell study proved that the formation of a supramolecular reservoir of ECN/TEP6 could achieve sustained drug release. Moreover, it was found that ECN/TEP6 statistically improved the healing effect compared to free ECN and M + ECN treatment in an *S. aureus*-infected mouse wound model. Considering the great market potential of topical preparations, we foresee convenient and biocompatible drug reservoirs, such as ECN/TEP6, may provide salutary benefits for patients. The synthesis of other liquid macrocycles and studies exploring their application are ongoing in our laboratory.

## Data Availability

Not applicable.

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
