# Peer review of "A Biocompatible Liquid Pillar[n]arene-Based Drug Reservoir for Topical Drug Delivery"

_pharmaceutics, 2022, doi:10.3390/pharmaceutics14122621_

Round 1
Reviewer 1 Report
In this research, authors described pillararenes-based drug delivery system for prolonged effects of antimicrobial drug, econazole nitrate (ECN). Oligoethylene oxide groups were introduced to prepare pillararenes derivatives and the authors screened out the optimal candidate (triethylene oxide substituents, TEP6) for sustained release of ECN with biocompatibility. In vivo studies confirmed that TEP6 forms supramolecular liquid reservoir and regulated release of ECN, contributing wound healing and antimicrobial effects. In overall, the manuscript is describing potential use of TEP6 for long-term topical drug delivery, but the reviewer recommends the consideration of this study after a revision.
Major comments:
- The reviewer suggests adding explanation on the rationale of using pillararenes for topical drug delivery, compared to conventional biomaterials including natural or synthetic polymers.
- Please provide references or mechanisms on how the addition of oligoethylene oxide substitution afford flow properties and biocompatibility. In addition, the author should discriminate tissue adhesion with maintenance of the treated materials by high viscosity. Regarding the explanation, please provide references on the relation between the shear viscosity and tissue adherence, or on the tissue adhesion of pillararenes..
- Add all the experimental methods in details. For example, the concentrations used in each experiments, detailed procedures of in vivo compatibility and disease model experiments (sampling time points, treatment methods (treatment numbers? topical spreading? and etc) are not clear.
- From the toxicity tests, it is clear that TEP5, QEP5, TEP6, and QEP6 show greater biocompatibility compared to DEP5, and DEP6. However, is all the concentration tested in cell-based assays can cover the range of concentration for in vivo experiment? In addition, to demonstrate in vivo compatibility the region of interest for H&E staining should be extended to dermis and hypodermis as the materials
- It is very important to address potential toxicity of the used material. Please provide information on the in vivo degradation or long-term toxicity of the polymers.
- Regarding the sustained release of the drug, it should be remained in the treated region for a long-term. How long can the treated materials be maintained? If the formulation is similar to conventional topical cream, so that the repeated treatment is required, the authors should state the importance of sustained release in this system.
- Is there any process of confirmation on the establishment of infection model? If the authors insist the improved wound healing by the reduced infection, please demonstrate any characteristic of infection in the control group.
- Figure 5b should be relocated in figure 6.
Reviewer 2 Report
The manuscript discusses the use of liquid pillar[n]arenes as a drug reservoir for topical drug delivery. The authors demonstrate that these materials can be used to deliver a variety of drugs, including hydrophobic and hydrophilic drugs, to the skin. The authors also show that the materials are biocompatible and do not cause any adverse effects on the skin. Overall, the manuscript is well-written and the data presented is clear and convincing. The use of liquid pillar[n]arenes as a drug reservoir is a novel approach and the data presented here demonstrates the potential of this approach for topical drug delivery. However, there are a few areas that could be improved.
1. the authors should provide more details on the methodology in terms of reference used. For example, the authors should provide suitable reference for the section 2.3. Biocompatibility evaluation of OEPns, section 2.6. In Vitro permeability studies and section 2.2. Rheological studies.
2. In the methodology, section 2.4. has been named as NMR spectroscopy while in the result section the result has been presented under subsection 3.4. “Host-guest recognition. Why the difference? Be specific for naming the subsection of methodology and result. There should not be variation otherwise it misleads the reader.
3. Similarly, in the methodology a separate subsection, 2.5. Fluorescence titration has been provided but the results has not been presented as separate section. Be specific.
4. Abbreviation section should be provided separately.
5. Grammatical errors and spellings should be checked thoroughly throughout the manuscript.
Round 2
Reviewer 1 Report
The authors have answered and revised all the comments raised by the reviewer.